# Surgical Resection and Immediate Reconstruction with a Bilayer Wound Collagen Matrix of a Rare Oral Angiosarcoma: A Case Report

**DOI:** 10.3390/diseases12060117

**Published:** 2024-06-03

**Authors:** Enzo Iacomino, Chiara Fratini, Laura Sollima, Alberto Eibenstein, Christian Barbato, Marco de Vincentiis, Antonio Minni, Federica Zoccali

**Affiliations:** 1Department of Life, Health, and Environmental Sciences, Università degli Studi di L’Aquila, 67100 L’Aquila, Italy; enzoiacomino74@hotmail.com; 2Department of Sense Organs, Sapienza University of Rome, Viale del Policlinico 155, 00161 Rome, Italy; chiara.fratini135@gmail.com (C.F.); marco.devincentiis@uniroma1.it (M.d.V.); federica.zoccali@uniroma1.it (F.Z.); 3Anatomy and Pathological Histology Division, ASL 1 Avezzano-Sulmona-L’Aquila, San Salvatore Hospital, 67100 L’Aquila, Italy; lausoll@hotmail.it; 4Department of Applied Clinical and Biotechnological Sciences, University of Aquila, 67100 L’Aquila, Italy; alberto.eibenstein@univaq.it; 5Institute of Biochemistry and Cell Biology (IBBC-CNR), Department of Sense Organs, Sapienza University Rome, Policlinico Umberto I, 00161 Roma, Italy; 6Division of Otolaryngology-Head and Neck Surgery, ASL Rieti-Sapienza University, Ospedale San Camillo de Lellis, 02100 Rieti, Italy

**Keywords:** oral angiosarcoma, surgery of vascular tumor, bilayer wound collagen matrix, skin regeneration technique

## Abstract

Angiosarcomas are malignant vascular tumors that commonly occur on the skin of the head and neck, breast, or scalp. Oral angiosarcoma is a rare tumor (0.0077% of all cancers in Europe), and regarding this atypical localization, no formal treatment trials have been conducted yet. We present a case of a 58-year-old female patient with a diagnosis of oral angiosarcoma. After tumor excision was performed by transoral surgical approach, immediate reconstruction of the intraoral surgical defects was made using Integra^®^ bilayer wound collagen matrix. A skin regeneration technique has previously been reported to provide good healing for defects of buccal resection, preventing postoperative cicatricial fibrosis.

## 1. Introduction

Although vascular endothelium is found extensively throughout the human body, malignant tumors originating from vaso-formative mesenchyme are exceptionally uncommon. Angiosarcoma is the prevailing terminology used to characterize these sarcomas originating from blood vessels. Angiosarcoma is a subtype of soft tissue sarcoma and is a rare, malignant type of cancer that develops in the lining of blood and lymph vessels. It falls within the broad category of vascular tumors. It has a similar distribution between sexes and can develop at any age, although the average age of occurrence among patients is typically over 70 years. This type of cancer can arise in any soft tissue structure or viscera. It most commonly affects the skin of the head and neck, the breast, or the scalp. If angiosarcoma affects different organs, it occurs in deeper tissues such as the liver or the heart [1,2]. The mouth and antrum are rare localizations for this tumor. Angiosarcomas constitute 2% of all soft tissue sarcomas [3], and 5–4% of cutaneous soft tissue sarcomas are angiosarcomas [4]. Primary malignant vascular tumors of the oropharynx are rare (0.0077% of all cancers in Europe), and the optimal management is still undefined [5,6,7,8]. The clinical presentation of angiosarcomas depends on the location and degree of tumor differentiation. The histological characteristics of these tumors vary greatly, ranging from well-differentiated tumors resembling hemangiomas to highly anaplastic variants resembling carcinomas or melanomas. Angiosarcoma of low grade exhibits a little solid portion with low-grade cellular characteristics and plentiful open vascular lumens. On the other hand, high-grade lesions are densely packed with cells and invasive, featuring a high mitotic rate and anomalous cells. Most cases are sporadic and due to radiation exposure, chemicals, or chronic lymphedema, and the risk of local recurrence is high. Oral angiosarcoma, primary or secondary, usually presents as a solitary, rapidly growing painless mass. Few symptoms are associated with the insidious onset of tumor growth. In the oral cavity, it appears as a wide-based, pliable, purple or violet mass, with blurred edges. The texture is lumpy, and in some areas, ulcers are present, likely caused by chewing trauma. Bleeding emerges as a serious late indicator and typically precedes extensive metastasis. Given the aggressively malignant character of the growth, swift invasion leads to pressure-related symptoms due to nearby structure involvement.

The overall 5-year survival rate is approximately 35%, even in the case of localized disease. The most optimistic survey rate suggests that only 60% of patients survive for more than 5 years [9,10,11]. There are no randomized clinical trials regarding the treatment of localized primary angiosarcomas; for this reason, the therapeutic choice depends mostly on the localization of the tumor [12]. In case of oral angiosarcoma localization, current treatment options are limited; ablative surgery is the main curative treatment option proposed, followed by radiotherapy and/or chemotherapy [13,14,15]. Radiation treatment alone is generally considered inadequate for potentially curable cases, and further radiotherapy is typically avoided for radiation-induced angiosarcomas. No formal and conclusive trials have been conducted yet, but if surgery is the first option for treatment, the problem lies in the reconstruction after surgery of the defects of the mucosa, which often contribute to a lower quality of life for the patient. After surgery, defects of the oral mucosa can lead to pain, infection, and undesirable healing, so adequate reconstruction is necessary and highly recommended in case of defects that occur after resection of oral lesions. There are many possibilities, including primary closure of intraoral defects that is often precluded by insufficient tissue availability, ranging from secondary healing to mucosal or, for large surgical resections, microvascular free flaps such as free radial forearm or anterolateral thigh flaps or skin grafts [16,17], which have become a highly successful reconstruction modality, but these involve complex surgical procedures and are associated with increased costs and concerns for donor site morbidities and infections.

We present a rare case of primary oropharyngeal angiosarcoma involving unusual and rare sites in a 58-year-old female. The patient underwent partial pharyngectomy, and an immediate reconstruction of intraoral surgical defects was performed by Integra^®^ bilayer wound collagen matrix (Integra Lifesciences, Princeton, NJ 08540, USA). The use of a skin regeneration technique has previously been reported to provide good healing for defects of buccal resection, preventing post-operative cicatricial fibrosis. In recent years, biosynthetic skin and mucosal substitutes for post-surgical reconstruction, including epidermal, dermal (Alloderm^®^, LifeCell, Branchburg, NJ, USA), and composite (Integra^®^, Integra Lifesciences, Princeton, NJ 08540, USA) grafts, have been used [18,19]. We decided to use a porous matrix of cross-linked bovine tendon collagen, glycosaminoglycan, and a semipermeable polysiloxane, a silicone layer structure Integra^®^ Bilayer Wound Matrix (Integra Lifesciences, Princeton, NJ 08540, USA), because the semipermeable silicone membrane modulates water vapor loss, provides a flexible adherent covering for the wound surface, and increases the device’s tear strength. Studies in the literature highlight that the use of the Integra template is non-allergenic and allows for complete tissue re-epithelialization after its removal; the biodegradable collagen–glycosaminoglycan matrix represents a scaffold for cells and capillaries [20,21,22].

## 2. Detailed Case Description

A 58-year-old Italian female presented to the ENT Unit, San Salvatore Hospital (L’Aquila), in February 2022 complaining of an intraoral vascular lesion over the left oral mucosa. The patient referred not having significant symptoms but a history of protracted bleeding from the gingiva related on two separate occasions, approximately 10 days before referral, and a sensation of foreign body during chewing for about 5–6 months. The patient reports no smoking habit and does not drink alcohol. She consumes two cups of coffee daily, follows a healthy diet, and is fit and well. The patient reports being claustrophobic and unable to undergo MRI scans. She has never undergone major surgery and has no allergies except for lactose intolerance. The patient has no significant chronic conditions, she does not take-home therapy, and she has never undergone radiotherapy or chemotherapy. She had not visited the dentist for at least 3 years and had never undergone an ear, nose, and throat examination (otorhinolaryngological evaluation). 

During intraoral examination, red lesions were observed on the lateral alveolar mucosa of the left quadrants, left retromolar trigone, lower left third molar region, and upper and lower left gingiva. These multifocal small polypoid reddish masses measured approximately 1 × 2 cm in dimension. There was associated bleeding upon palpation, and the masses had a soft to firm consistency. No other relevant clinical findings were noted. Flexible laryngoscopy did not reveal any other localizations of disease. The patient, presenting with the chief complaint of painless, red, ulcerated, friable, and progressively enlarging masses, referred that the lesions had been present for approximately six months. The past medical history was non-contributory, and assessment of the head and neck region revealed no lateral cervical lymph node enlargement. After a few days, a biopsy of the mass on the left lateral alveolar mucosa was performed, as shown in Figure 1. 

The specimens were fixed in 10% neutral buffered formalin and processed routinely for paraffin sections. Sections were cut and stained with hematoxylin and eosin for light microscopy examination. Histologically, the tumor was composed of irregular vascular channels and spindle cells. Furthermore, solid areas consisting of spindle cells with variable nuclear atypia and mitotic activity were found, as well as vascular channels with membranous expression of CD31 immunostaining, as shown in Figure 2.

Based on the clinical features, microscopic findings, and histopathological examination, a diagnosis of angiosarcoma of the oral cavity was performed. Total body computed tomography (TB CT) imaging revealed a small soft tissue density lesion involving the upper left lateral alveolar mucosa. The lesion appeared as an inhomogeneous enhanced 2 cm ovoid mass with polycyclic borders (indicated by the white arrow). There was no evidence of generalized cervical lymph node enlargement or other localizations of angiosarcoma (Figure 3). 

Local disease radical surgery with complete resection (R0) was planned as the primary treatment of choice. On 31 May 2022, tumor excision was performed through a transoral approach. A mucosa and submucosa incision were made up to the muscular level of the left gingival region, extending upwards to the tuber maxillae, downwards to the retromolar trigone, posteriorly to the anterior tonsillar pillar, and superiorly to the Stenon duct. The entire area, including the lesions, was removed en bloc, preserving the lingual nerve (Figure 4).

Both the intraoperative and definitive examination of margins showed negative histological results. Immediate repair of the buccal defect resulting from the cancer resection was performed during the same surgical session. The reconstruction used an appropriately sized oval bilayer dermal regeneration template (Integra^®^ Dermal Regeneration Template, Integra Lifesciences, Princeton, NJ 08540, USA) (Figure 5). 

The template was secured correctly with 3-0 Vicryl sutures to the mucosa, ensuring proper anchorage at the edge of the defect [23,24]. After the surgery, the patient was closely monitored with follow-up observation every two days during the first week to check for any signs of infection, dehiscence (wound opening), or loss of the dermal regeneration template. After 20 days, morpho-functional outcomes were analyzed, and it was observed that the patient did not show any infections or loss of the dermis. The silicon sheet for the reconstruction was detached and complete re-epithelialization of the lesion was highlighted by the granulation tissue underneath. At 8 weeks post-operative, it was determined that the defect healed completely without further reconstruction. Clinically and histologically connective fibrous tissue was observed and detected (Figure 6 and Figure 7). 

## 3. Discussion

Angiosarcomas are malignant vascular tumors, and approximately 60% of them occur in the head and neck region, with the scalp being the most frequent site, followed by the face and the neck. Angiosarcoma of the oral cavity is rare, and an optimal management protocol treatment for this tumor remains undefined. Oral angiosarcoma typically presents as a rapidly growing, painless, and bleeding mass. It appears as a polypoid nodular bluish mass, and minimal symptoms are associated with the insidious onset of tumor growth. Oral angiosarcoma has a great propensity to mimic benign inflammatory diseases such as chronic periodontal disease or pyogenic granuloma. Other frequent erroneous oral diagnoses are hemangioma, hematoma, or giant cell epulis. Benign disease is the original (erroneous) diagnosis in as many as 75% of cases, but a vascular origin, albeit granulation tissue, is usually apparent from the start. The histological features of angiosarcoma can vary both within and between cases, morphological differences can be subtle, and distinguishing a malignant vascular tumor from a benign proliferative or inflammatory lesion with light microscopy can be difficult.

In our case, a 58-year-old female patient presented oral polypoid reddish masses in the left oral mucosa for about six months. Diagnosis of angiosarcoma in oral cavity lesions was a complex process due to the uncommon age, unusual site, and significant clinical consequences associated with this finding, and this did not lead to a delay in therapy in our case. Furthermore, the lack of precise treatment guidelines and limited data on the tumor’s clinical and pathological features and prognostic factors made management decisions a challenge. After confirming the diagnosis through histological examination, including immunohistochemistry to confirm the vascular phenotype [25,26] and ruling out tumors in other parts of the tissue through TB CT, our patient underwent partial pharyngectomy, and local resection surgery was chosen as the primary treatment. 

Involved margins (R1 or R2 resection) are common because of the invasive and often multifocal nature of angiosarcomas, and the surgical goal was to achieve complete tumor resection with histologically negative and adequate resection margins while preserving the lingual nerve and as much healthy tissue as possible to minimize functional loss [22,23,24,25,26,27,28]. Angiosarcomas often involve margins due to their invasive and multifocal nature [29]. During the same surgical session, the second step involved the reconstruction process. Several options, such as free flaps, microvascular flaps, or skin grafts, could have been considered. However, after reviewing the literature, no specific free flap [30,31] or skin graft appeared superior, and all of them could have increased patient morbidity. Therefore, we opted for reconstruction using the Integra^®^ Dermal Regeneration Template, an advanced skin/mucosa replacement system placed directly on the excised wound. The Integra^®^ Dermal Regeneration Template is a bovine collagen/glycosaminoglycan dermal replacement covered by a silicone temporary epidermal substitute. Integra^®^ is a blend of bovine type I collagen and shark chondroitin-6-sulfate glycosaminoglycan combined with a silicone pseudo epidermis. Bovine dermal collagen promotes the integration of fibroblasts into the Integra structure, leading to the development of a new layer of skin (neo-dermis). This favors the correct positioning of a graft on the neo-dermis. Some studies have demonstrated the effectiveness of a one-step Integra^®^ procedure followed by natural healing in diverse head and neck regions. It provides a three-dimensional porous matrix that acts as a scaffold for cell migration [31]. In addition, the cost of the Integra^®^ bilayer wound matrix is higher than that of an autologous skin graft; it can be offset by the reduction in intraoperative time and equipment required for graft harvesting, as well as postoperative management of the donor site [32]. It provided immediate wound closure and positively influenced the patient’s quality of life by reducing morbidity and infections and preserving swallowing, speech, and normal oral functions. The use of this skin regeneration system proved to be highly effective. After 20 days from surgery, when the silicon sheet of the Integra matrix was detached, successful early granulation tissue was clinically observed. After 8 weeks, histological examination revealed the regrowth and differentiation of connective fibrous tissue with scan fibroblasts. After 4 months, the patient underwent a medical evaluation and was reported as not having significant symptoms or pain, except for a progressive limitation of the oral opening; for this reason, solid food was reduced in her diet. Oral examination showed well-healed oral mucosa lined the surgical defect with complete closure of the oral defect. Mouth opening was more difficult compared to the first postoperative evaluation. The scar tissue created marginal tensions that prevented a complete mouth opening. Thanks to our experience, the limitation of the oral opening was managed conservatively by performing progressive and continuous distraction of the scar tissues with an external distractor. The patient was instructed to perform the ‘distraction’ at home, by applying two rounds a day of distraction treatment for 30 min with the external distractor for a month. We conducted a medical examination 30 days after the initial evaluation, and an approximately normal oral opening was achieved [32]. The patient underwent a third medical evaluation after one year from surgery; she referred no symptoms, no pain, and no more limitation of oral opening, and she did a total-body CT scan evaluation with negative results. No recurrence of the disease or suspicious areas were detected, and the one-year oncological outcome was good, with local recurrence-free survival and distant metastasis-free survival. 

## 4. Conclusions

The epidemiological evaluation further confirms the rarity of oral cavity involvement with angiosarcoma. Therefore, it is crucial to conduct an early diagnosis, and a definitive diagnosis through biopsy is imperative after attempting to alleviate local causative factors. For these reasons, our early involvement in oral diagnosis and biopsy of this lesion improves the prognosis for the patient’s survival. This presentation will reinforce the aggressive pursuit of seemingly benign oral lesions.

Considering that oral angiosarcoma is a rare localization, doing a surgery excision and defect reconstruction using the Integra^®^ template proved to be a highly effective treatment option: simple and representative. It offers a successful alternative for soft tissue reconstructions, as it allows for easier handling, restoration of oral functions, promotion of hemostasis, pain relief, and induction of granulation. The procedure is minimally invasive, results in good healing, and offers the advantages of reproducible tissue engineering technologies; these mucosal substitutes/oral mucosal equivalents can be used for tissue repair and represent a significant advantage in oral reconstructive surgery.

Having demonstrated the effectiveness of this method of single-step skin regeneration, followed in this case by granulation and healing by secondary intention, we suggest that it could also be useful in comparable scenarios involving mild to moderate intraoral defects of different origins. Immediate reconstruction surgery after cancer resection could significantly contribute to a more effective repair of oral mucosa after surgical excision and faster oral rehabilitation and could influence a patient’s quality of life.

## Figures and Tables

**Figure 1 diseases-12-00117-f001:**
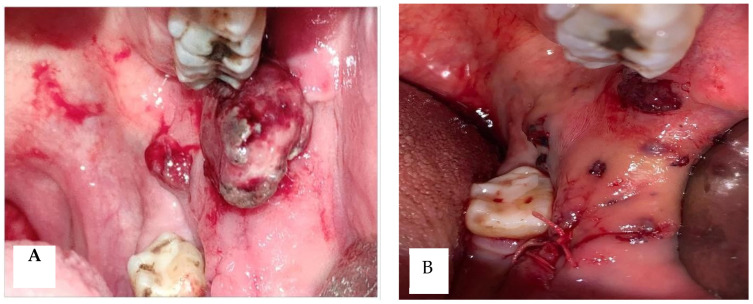
Clinical and imaging findings of intraoral angiosarcoma before biopsy (**A**); clinical and imaging findings of intraoral angiosarcoma after biopsy (**B**).

**Figure 2 diseases-12-00117-f002:**
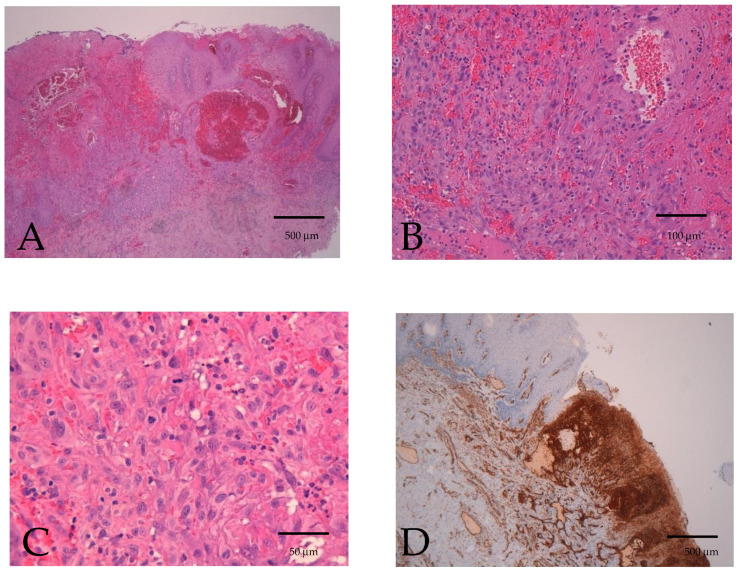
Biopsy specimen. Hematoxylin and eosin 4×, epithelial ulceration, and sub-epithelial ill-defined tumor composed of irregular vascular channels and spindle cells. Marked surrounding inflammatory infiltrate (**A**). Hematoxylin and eosin 20×, solid area composed of spindle cells with variable nuclear atypia and hemorrhagic extravasations (**B**). Hematoxylin and eosin 40×, solid area composed of spindle cells with variable nuclear atypia and mitotic activity (**C**). CD31 4×, vascular channels with membranous expression of CD31 immunostaining (**D**).

**Figure 3 diseases-12-00117-f003:**
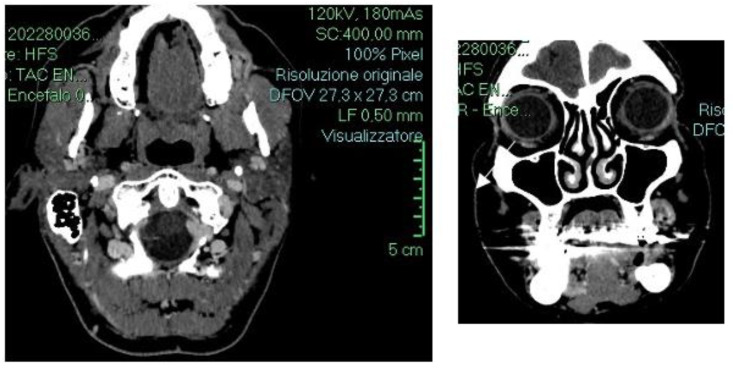
Axial and coronal CT images showing the Angiosarcoma.

**Figure 4 diseases-12-00117-f004:**
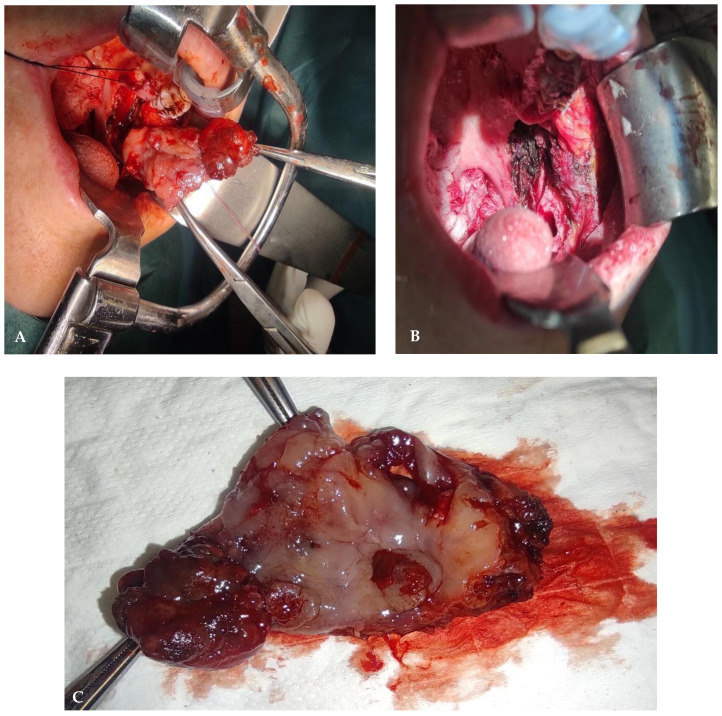
Intraoperative time of tumor excision (**A**). Surgical oral defect following tumor excision (**B**). Tumor specimen removed en bloc (**C**).

**Figure 5 diseases-12-00117-f005:**
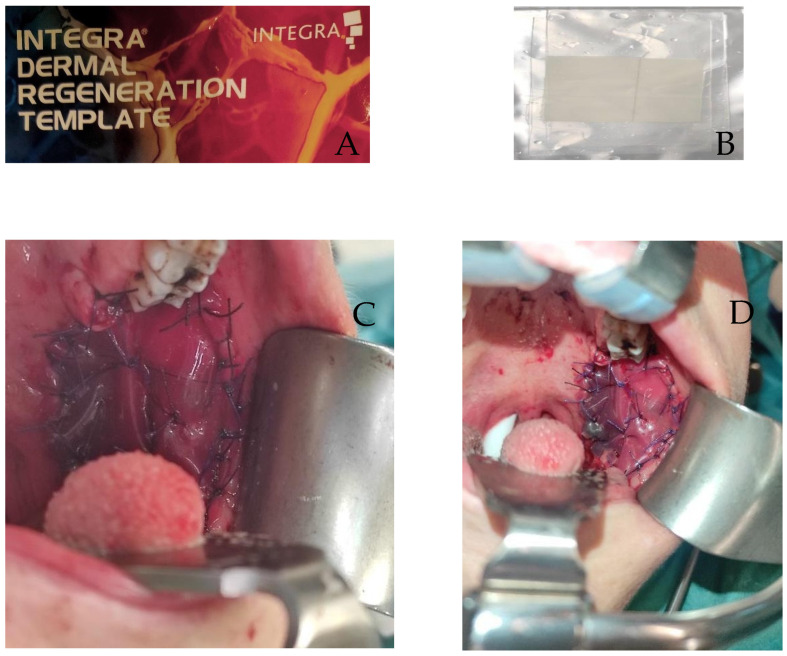
Integra^®^ Dermal Regeneration Template, Integra LifeSciences (**A**), silicone layer of a porous matrix of cross-linked bovine tendon collagen, glycosaminoglycan, and a semipermeable polysiloxane (**B**). The created defects are covered with an Integra bilayer secured with 3-0 Vicryl sutures to the oral mucosa (**C**,**D**).

**Figure 6 diseases-12-00117-f006:**
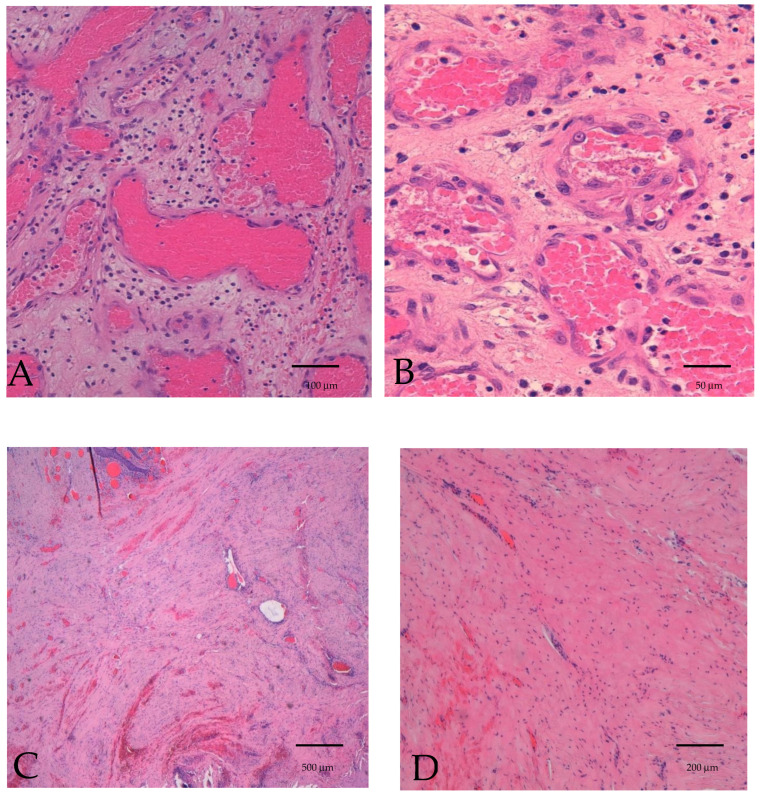
Surgical specimen: irregular vascular channels lined by hyperchromatic moderately atypical vascular endothelial cells (**A**,**B**). Hematoxylin and eosin 20× (**A**). Hematoxylin and eosin 40× (**B**). Hematoxylin and eosin 4× postoperative submucosal cicatricial fibrosis with mild hemorrhagic extravasation (**C**). Hematoxylin and eosin 10× postoperative cicatricial fibrosis: connective fibrous tissue with scant fibroblast (**D**).

**Figure 7 diseases-12-00117-f007:**
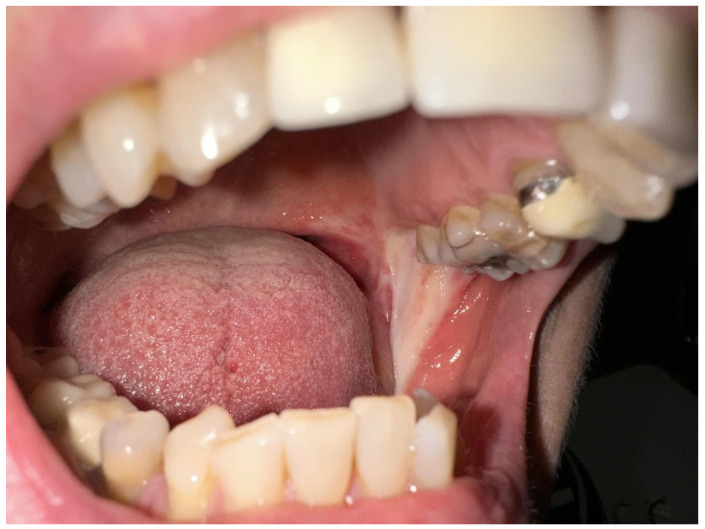
Eight weeks post-operative cicatricial fibrosis of oral mucosa.

## Data Availability

Data are contained within the article.

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
