# Peer review of "Surgical Resection and Immediate Reconstruction with a Bilayer Wound Collagen Matrix of a Rare Oral Angiosarcoma: A Case Report"

_diseases, 2024, doi:10.3390/diseases12060117_

Round 1
Reviewer 1 Report
Comments and Suggestions for Authors
This case report describes the usefulness of a bilayer collagen matrix as a reconstructive material after surgical resection of oral angiosarcoma. It is well presented, including pathological findings, and is considered to meet a certain standard as a scientific case report.
On the other hand, what about the long-term outcome? It would be desirable to include whether there were any long-term problems with the wound and the oncological outcome.
Author Response
May 15, 2024
Dear Editors and Reviewers:
Re: Revised version of the manuscript Diseases-2991227
We express our gratitude to the editors and three reviewers for their valuable and insightful feedback on our original submission. We have carefully considered each comment and suggestion, and as a result, we have made significant revisions to our article to address the reviewers' concerns. We have indicated in red the modifications made in the revised version, addressing the reviewers' specific comments. Additionally, we have included some new information in the revised manuscript (also indicated in red). We hope that these changes have improved the manuscript and that the reviewers will find it suitable for publication in the Diseases journal.
Reviewer #1
Comments and Suggestions for Authors
This case report describes the usefulness of a bilayer collagen matrix as a reconstructive material after surgical resection of oral angiosarcoma. It is well presented, including pathological findings, and is considered to meet a certain standard as a scientific case report.
On the other hand, what about the long-term outcome? It would be desirable to include whether there were any long-term problems with the wound and the oncological outcome
R1) what about the long-term outcome? A concise answer is: “After 8 weeks, histological examination revealed the regrowth and differentiation of connective fibrous tissue with scan fibroblasts. After 4 months oral opening limitation was referred, and it was managed conservatively by performing progressive and continuous distraction of the scar tissues with an external distractor. One-year after surgery, the oncological outcome was good TC TB was negative and with local recurrence-free survival and distant metastasis-free survival”.
In the manuscript, we added the following sentences:
Line 274-291: After 20 days from surgery, when the silicon sheet of the Integra matrix was detached, successful early granulation tissue was clinically observed. After 8 weeks, histological examination revealed the regrowth and differentiation of connective fibrous tissue with scan fibroblasts. After 4 months, the patient underwent a medical evaluation and was reported not having significant symptoms or pain, except for a progressive limitation of the oral opening for this reason, solid food was reduced in her diet. Oral examination showed well-healed oral mucosa lined the surgical defect with complete closure of the oral defect. The mouth opening was more difficult compared to the first postoperative evaluation. The scar tissue created marginal tensions that prevented a complete mouth opening. Thanks to our experience, the limitation of the oral opening was managed conservatively by performing progressive and continuous distraction of the scar tissues with an external distractor. The patient was instructed to perform the ‘distraction’ at home, by applying two rounds a day distraction treatment for 30 minutes, two times a day with the external distractor for a month. We conducted a medical examination 30 days after the initial evaluation, and an approximately normal oral opening was achieved. [32] The patient underwent a third medical evaluation after one year from surgery, she referred no symptoms, no pain, and any more limitation of oral opening, and she did a total-body CT scan evaluation with negative results. No recurrence of the disease or suspicious areas were detected, and the one-year oncological outcome was good, with local recurrence-free survival and distant metastasis-free survival.
The authors would like to express their gratitude to Reviewer 1 for the insightful feedback. Your feedback on the communication of these potentially significant findings will contribute to enhancing the clarity and impact of our research.
Sincerely
Dr. Christian Barbato and Prof. Antonio Minni
Reviewer 2 Report
Comments and Suggestions for Authors
The article is a case report of a rare malignant pathology. The novelty is represented by the material used in the reconstruction. It could be helpful to mention the fundamental research that was based on this strategy. How was the acceptance of this material? Allergic or intollerance?
Has the patient had radiotherapy or chemotherapy after surgery? What was the impact of these treatments locally?
Author Response
May 15, 2024
Dear Editors and Reviewers:
Re: Revised version of the manuscript Diseases-2991227
We express our gratitude to the editors and three reviewers for their valuable and insightful feedback on our original submission. We have carefully considered each comment and suggestion, and as a result, we have made significant revisions to our article to address the reviewers' concerns. We have indicated in red the modifications made in the revised version, addressing the reviewers' specific comments. Additionally, we have included some new information in the revised manuscript (also indicated in red). We hope that these changes have improved the manuscript and that the reviewers will find it suitable for publication in the Diseases journal.
Reviewer #2
Comments and Suggestions for Authors
The article is a case report of a rare malignant pathology. The novelty is represented by the material used in the reconstruction. It could be helpful to mention the fundamental research that was based on this strategy. How was the acceptance of this material? Allergic or intollerance?
Has the patient had radiotherapy or chemotherapy after surgery? What was the impact of these treatments locally?
- a) We added sentences Lines 87-93: “We decided to use a porous matrix of cross‐linked bovine tendon collagen, glycosaminoglycan, and a semipermeable polysiloxane, a silicone layer structures Integra® Bilayer Wound Matrix (Integra®) because the semipermeable silicone membrane modulates water vapor loss, provides a flexible adherent covering for the wound surface, and increases the device's tear strength. Studies in the literature highlight that the use of the Integra template is non-allergenic and allows for complete tissue re-epithelialization after its removal; the biodegradable collagen-glycosaminoglycan matrix represents a scaffold for cells and capillaries [20,21,22].”
- b) Lines 99-104: The patient reports no smoking habit and does not drink alcohol. She consumes two cups of coffee per die and follows a healthy diet, she is fit and well. The patient reports being claustrophobic and unable to undergo MRI scans. She has never undergone major surgery and has no allergies except for lactose intolerance. The patient has no significant chronic conditions, she does not take-home therapy, and she has never undergone radiotherapy or chemotherapy.
The authors would like to express their gratitude to Reviewer 2 for the insightful feedback. Your feedback on the communication of these potentially significant findings will contribute to enhancing the clarity and impact of our research.
Sincerely
Dr. Christian Barbato and Prof. Antonio Minni
Reviewer 3 Report
Comments and Suggestions for Authors
Comments for this case report manuscript:
1. Does the patient have smoking and alcohol history?
2. Please add scale bar on the Figure 2 images.
3. What is blinker on Figure 3? Is it the same as arrow? What is the blue line on Figure 3?
4. More details needed for Figure 5 legend.
5. For figure 6: needs scale bars. the images are not identical in size. More details needed for figure 6 legend.
6. For Figure 7: use "eight weeks" is better than "8 weeks" because "8" is right after "7" so easy to get confused.
Author Response
May 15, 2024
Dear Editors and Reviewers:
Re: Revised version of the manuscript Diseases-2991227
We express our gratitude to the editors and three reviewers for their valuable and insightful feedback on our original submission. We have carefully considered each comment and suggestion, and as a result, we have made significant revisions to our article to address the reviewers' concerns. We have indicated in red the modifications made in the revised version, addressing the reviewers' specific comments. Additionally, we have included some new information in the revised manuscript (also indicated in red). We hope that these changes have improved the manuscript and that the reviewers will find it suitable for publication in the Diseases journal.
Reviewer #3
Comments and Suggestions for Authors
Comments for this case report manuscript:
- Does the patient have smoking and alcohol history?
- No, she doesn’t. The patient reports no smoking habit and does not drink alcohol. She consumes two cups of coffee per day and follows a healthy diet, she is fit and well, taking no regular medication.
- Please add scale bar on the Figure 2 images.
- All dimensions are reported in the figure legend.
- What is blinker on Figure 3? Is it the same as arrow? What is the blue line on Figure 3?
- In Figures 3 A and B the withe arrow indicates the lesion that appeared as an inhomogeneous enhanced 2-cm ovoid mass with polycyclic borders. The blue line in Figure 3 is an artifact. We take other images without artifacts.
- More details needed for Figure 5 legend.
4.The Figure 5 legend was rewritten as follows: “Integra® Dermal Regeneration Template, Integra LifeSciences (A), silicone layer of a porous matrix of cross‐linked bovine tendon collagen, glycosaminoglycan, and a semipermeable polysiloxane (B) The created defects is covered with a Integra bilayer secured with 3-0 Vicryl sutures to the oral mucosa (C, D).”
- For figure 6: needs scale bars. the images are not identical in size. More details needed for figure 6 legend.
- The Fig. 6 legend reported the size of the surgical specimen: Irregular vascular channels lined by hyperchromatic moderately atypical vascular endothelial cells (A,B)Hematoxylin and eosin 20x (A) Hematoxylin and eosin 40x (B). Hematoxylin and eosin 4x postoperative submucosal cicatricial fibrosis with mild haemorragic extravasation (C), Hematoxylin and eosin 10x postoperative cicatricial fibrosis: connective fibrous tissue with scant fibroblast.(D)
- For Figure 7: use "eight weeks" is better than "8 weeks" because "8" is right after "7" so easy to get confused.
- We replaced these terms.
The authors value your constructive feedback, which will undoubtedly contribute to the refinement and future direction of our research efforts.
Sincerely
Dr. Christian Barbato and Prof. Antonio Minni
Round 2
Reviewer 3 Report
Comments and Suggestions for Authors
The authors answered my questions and comments except adding the scale bar directly on the images of Figure 2 and 6.
Author Response
Dear Reviewer,
as requested we added the scale bar directly on the images of Figure 2 and 6.
Thank you very much for your collaboration,
kind regards
